The real Bigfoot: a pes from Wyoming, USA is the largest sauropod pes ever reported and the northern-most occurrence of brachiosaurids in the Upper Jurassic Morrison Formation

Maltese Anthony Anthony.Maltese@gmail.com 1
Tschopp Emanuel tschopp.e@gmail.com etschopp@amnh.org 2 3 4
Holwerda Femke 3 5 6
Burnham David 7
1 Rocky Mountain Dinosaur Resource Center , Woodland Park , CO , United States of America
2 Division of Paleontology, American Museum of Natural History , New York , NY , United States of America
3 GeoBioTec, Faculdade de Ciências e Tecnologia, Universidade Nova de Lisboa , Caparica , Portugal
4 Museu da Lourinhã , Lourinhã , Portugal
5 Bayerische Staatssammlung für Paläontologie und Geologie, Staatliche Naturwissenschaftliche Sammlungen Bayerns (SNSB) , München , Germany
6 Department of Earth Sciences, Utrecht University , Utrecht , Netherlands
7 Biodiversity Institute, University of Kansas , Lawrence , KS , United States of America
Wedel Mathew
Electronic publication date: 2018 Jul 24
Publication date: 2018
Volume: 6
Electronic Location ID: e5250
Received 2018 Mar 16; Accepted 2018 Jun 25
Copyright: ©2018 Maltese et al.
Copyright year: 2018
Copyright holder: Maltese et al.
License: This is an open access article distributed under the terms of the Creative Commons Attribution License, which permits unrestricted use, distribution, reproduction and adaptation in any medium and for any purpose provided that it is properly attributed. For attribution, the original author(s), title, publication source (PeerJ) and either DOI or URL of the article must be cited.
License URL: https://creativecommons.org/licenses/by/4.0/

Keywords: Jurassic, Morrison Formation, Titanosauriformes, North America, Pes, Brachiosauridae

Funding: Europasaurus-Projekt Theodore Roosevelt Memorial Fund Richard Gilder Graduate School at the American Museum of Natural History Funding for the collection visits was received by E. Tschopp through a Volkswagen-Stiftung fellowship within the “Europasaurus-Projekt”. Tschopp is currently holding a Theodore Roosevelt Memorial Fund and Division of Paleontology Postdoctoral Fellowship of the Richard Gilder Graduate School at the American Museum of Natural History, New York. The funders had no role in study design, data collection and analysis, decision to publish, or preparation of the manuscript.

==============================
A set of associated left pedal elements of a sauropod dinosaur from the Upper Jurassic Morrison Formation in Weston County, Wyoming, is described here. Several camarasaurids, a nearly complete small brachiosaur, and a small diplodocid have been found at this locality, but none match the exceptionally large size of the pedal elements. Next to the associated pedal elements, an isolated astragalus, phalanx and ungual were found, which match the large metatarsals in size. The elements cannot be ascribed to diplodocids due to the lack of a ventral process of metatarsal I. Moreover, the morphology of metatarsal V has a broad proximal end, with a long and narrow distal shaft, which differs from Camarasaurus. The size of the material and a medially beveled distal articular surface of metatarsal IV imply an identification as a brachiosaurid. This is the largest pes ever reported from a sauropod dinosaur and represents the first confirmed pedal brachiosaur elements from the Late Jurassic of North America. Furthermore, this brachiosaur material (the pes and the small nearly complete specimen) is the northernmost occurrence of brachiosaurids in the Morrison Formation.

Introduction

The Upper Jurassic (late Oxfordian to early Tithonian) Morrison Formation is famous for its abundant dinosaur material, particularly sauropods (e.g., Camarasaurus, Diplodocus, Apatosaurus, and Brachiosaurus; McIntosh, 1990a; McIntosh, 1990b; Foster, 2003; Chure et al., 2006; Whitlock, 2011; Woodruff & Foster, 2017; Tschopp & Mateus, 2017). Occurrences of these sauropods are recorded throughout the Morrison Formation, which outcrops in eight states, but it remains unclear if the more than 20 known species co-occurred in the same place or if they were segregated geographically. This is particularly true for species that are rarely found, such as Dyslocosaurus polyonychius (McIntosh, Coombs & Russell, 1992), Dystrophaeus viaemalae (Cope, 1877; McIntosh, 1997), Suuwassea emilieae (Harris & Dodson, 2004), Kaatedocus siberi (Tschopp & Mateus, 2013), and Brachiosaurus altithorax (Riggs, 1903; Riggs, 1904; Bonnan & Wedel, 2004).

The northern exposures of the Morrison Formation are little-known compared to the ones farther south. In order to test for geographical segregation among sauropods, it is therefore crucial to assess the taxonomy of any specimen found in the north in as much detail as possible, no matter how incomplete the specimens are. Herein, we describe a partial, potentially brachiosaurid pes from the Black Hills in Wyoming. Pedal elements can be diagnostic at least at family level, sometimes even below that (McIntosh, Coombs & Russell, 1992; D’Emic, 2012; Mannion et al., 2013; Tschopp et al., 2015). Though found together with Camarasaurus, there are morphological differences that show the new foot to be dissimilar to both Camarasaurus specimens from this quarry. Brachiosaurid material from this site has been reported in the past (Foster, 2003; Bader, Hasiotis & Martin, 2009), but without a detailed systematic assessment or description. Given that these would be the northern-most occurrences of brachiosaurids in the Morrison Formation, the herein described pes adds important data to our understanding of geographical patterning of the Morrison Formation fauna.

Materials and Methods

Material and association

The pes described herein consists of an astragalus (KUVP 142200), metatarsals I to V, four non-ungual pedal phalanges, one ungual (KUVP 129724), an additional non-ungual phalanx (KUVP 133862), and a second ungual (KUVP 144767). It was found at the Bobcat Pit site in Weston County in the Black Hills in north-eastern Wyoming (see Fig. 1A). It has been mentioned in Bader, Hasiotis & Martin (2009), but never described in detail.

Figure 1 Location (A) and quarry maps (B, C) of Bobcat Pit in Weston County, Wyoming.

The astragalus and pes described herein (KUVP 129724, 133862, 142200, 144767) were found associated with the Camarasaurus skeletons KUVP 129713 and 129716. Quarry maps modified from Bader, Hasiotis & Martin, 2009: figs 2, 4.

In addition to the elements belonging to the pes described herein (KUVP 129724, 133862, 142200, 144767, Figs. 1B and 1C), Bobcat Pit has produced several specimens belonging to camarasaurid, diplodocid, and brachiosaurid sauropods (Bader, Hasiotis & Martin, 2009). During a 1998 expedition led by the University of Kansas, the sauropod pes was found underneath the tail of the Camarasaurus KUVP 129716, with the phalanges scattered around the skeleton (Fig. 1C). Metatarsals I, II, III, and IV of KUVP 129724 were closely associated, whereas metatarsal V and a pedal ungual (likely from digit III) were found nearby. Three proximal phalanges (field numbers BP013, BP194 and BP208; see Table 1) were recovered about a meter away from the metatarsals with a proximal phalanx (field number BP185) slightly further away. Phalanx KUVP 133862 was discovered during preparation of a large field jacket containing caudal elements of Camarasaurus KUVP 129716. The astragalus KUVP 142200 was collected beneath KUVP 129713. A second large claw, likely php I-2, was discovered when the site was later reopened by another excavation crew. This claw was molded and a high fidelity cast was donated to KUVP, bearing the number KUVP 144767. All elements described herein are referred to the same animal as KUVP 129724 due to their great size, relative proximity in the quarry, and lack of any duplication in the elements.

Table 1 Measurements of brachiosaurid pes elements from Bobcat Pit (in mm).

Catalog numbers are indicated for the elements not included in KUVP 129724.

Element	Length	Proximal width	Distal width	Field number	
Astragalus (KUVP 142200)	246	370		–	
mt I	266	133	167	BP099	
mt II	290	163	183	BP098	
mt III	332	134*	156	BP097	
mt IV	329	154	134*	BP145	
mt V	269	182	91	BP096	
php I-1	101	132	102	BP208	
php II-1	100	147	130	BP013	
php III-1	81	135	123	BP194	
php ?IV-1	80	99	105	BP185	
php ?V-1 (KUVP 133862)	52	68		–	
Ungual ?III	185	52		BP014	
Notes.

* Asterisks mark widths as preserved in elements with damaged bone surfaces.

Abbreviationsmt metatarsal

php pedal phalanx

Based on comparisons with articulated camarasaurid and brachiosaurid pedes, we interpret the phalanges as php I-1, II-1, III-1, and possible IV-1 and V-1, and the unguals as probably representing unguals I and III. However, given that the specimen was found disarticulated and incomplete, we refrain from reconstructing a pedal formula.

The elements of KUVP 129724, the astragalus KUVP 142200, the phalanx KUVP 133862, and the ungual KUVP 144767 were not consistent in size with the Camarasaurus specimen they were found with (KUVP 129716), nor with a second, larger Camarasaurus specimen from the same quarry. The Camarasaurus KUVP 129716 was nearly complete and included almost all the pedal material in articulation. All pedal bones from this specimen are duplicated in KUVP 129724, so it is certain the large pes does not belong to this specimen. A larger Camarasaurus (KUVP 129713) was excavated in 1997, approximately 7m adjacent in the same quarry. However, this individual is also much smaller than the new pes. Finally, all proximal phalanges display a peculiar bone texture on their proximal articular surfaces. These surfaces are marked by irregularly undulating grooves generally extending from the margins towards the center. Such a texture is likely due to remodeling in response to specific stresses in vivo, supporting the interpretation that all phalanges belong to a single pes, because all the joints between metatarsals and phalanges seem to be equally affected. As specimens at this locality generally occur as discreet skeletons rather than a mass of bonebed elements, these considerations suggest it is very likely the pes is a slightly scattered assemblage of elements from a single individual.

Excavation and preparation

The pes and astragalus were excavated from a mudstone deposit, with some encrustation of caliche on the bones, especially around the articular ends. The softer matrix was removed primarily with X-acto knives and air abrasion utilizing sodium bicarbonate abrasives. Concretionary material was removed much more slowly employing Aro and Chicago Pneumatic air scribes and air abrasion with Dolomite (and very seldom glass beads and Aluminum Oxide) abrasives. All elements were scanned using an Artec Spider handheld structured light unit and processed using Artec Studio 12 software. Individual scan files were organized and arranged in Blender software to produce figure images. The three-dimensional models are available through KUVP for research purposes.

Description and Comparison

Astragalus

The astragalus KUVP 142200 (Fig. 2) is slightly wider transversely than proximodistally tall and anteroposteriorly long (Table 1). It has neosauropod affinities based on the ascending process that reaches the posterior margin (Wilson & Sereno, 1998). As in most sauropods, it is wedge-shaped, with a reduced medial corner. However, it differs from diplodocids and camarasaurids by a more pentagonal instead of subtriangular outline in posterior view (Fig. 3). The extension of the medial corner is similar to the brachiosaurids Giraffatitan and Lusotitan, which have a relatively shorter and more rounded medial end than Janenschia and Camarasaurus (Fig. 3; Janensch, 1961; Mannion et al., 2013; Tschopp et al., 2015). The lateral surface of the astragalus KUVP 142200 received the fibula. It faces laterally, and has no distinct bony shelf that would have supported the fibula, unlike the condition in diplodocids (Whitlock, 2011; Tschopp, Mateus & Benson, 2015).

Figure 2 Single bones of the brachiosaurid pes described herein.

Astragalus KUVP 142200 in proximal, distal, anterior, posterior, medial and lateral view, and metatarsals I to V, phalanges I-1 to IV-1 (KUVP 129724), phalanx V-1 (KUVP 133862), and unguals I (KUVP 144767) and III (KUVP 129724) in plantar, lateral, dorsal, medial, proximal and distal views. Dorsal surface in proximal and distal views points upwards. Scale bar = 10 cm (valid for all bones).

Figure 3 Comparative outline drawings of neosauropod astragali in posterior view.

KUVP 142200 (left) is compared to the brachiosaurids Giraffatitan (MB.R.2562, left; traced from Janensch (1961)), the camarasaurid Camarasaurus (AMNH FARB 5761, right reversed; traced from Osborn & Mook (1921)), and the diplodocid Galeamopus (SMA 0011, left; traced from Tschopp & Mateus (2017)). Note the expanded shelf with a distinctly convex margin below the fibular facet in the diplodocid Galeamopus (grey arrow). Drawings scaled to equal transverse width in order to highlight shape differences.

Metatarsals

The pes KUVP 129724 (Fig. 2) has the typical shape of a eusauropod pes, having a spreading, asymmetrical metatarsus with an entaxonic structure, where mt I is the most robust element (Table 1; Coombs Jr, 1975; Cooper, 1984; McIntosh, 1990a; Farlow, 1992; Upchurch, 1998; Wilson & Sereno, 1998; Bonnan, 2005).

The metatarsals (Fig. 2) are generally hour-glass shaped with transversely and dorsoplantarly expanded proximal and distal articular surfaces. As is typical for eusauropods, the mt V differs from the rest in having a much more widely expanded proximal end compared to the distal one, resulting in a paddle-like shape (Bonnan, 2005). The distal articular surfaces bear distinct condyles in mt I, which gradually decrease in size and distinctiveness towards mt V with its gently rounded surface without any differentiation into separate condyles.

Figure 4 Comparative outline drawings of neosauropod metatarsals I in proximal (A–C) and dorsal view (D–F).

KUVP 129724 (A, D; left metatarsal) is compared with the brachiosaurid Vouivria MNHN.F.1934.6 DAM 12 (B, E; left metatarsal; traced from Mannion, Allain & Moine, 2017) and the flagellicaudatan Galeamopus SMA 0011 (C, F; left metatarsal; traced from Tschopp & Mateus, 2017). Note the pointed dorsolateral corner of the proximal articular surface in the brachiosaurids (arrows). Drawings scaled to equal transverse width (A–C) and proximodistal length (D–F) in order to highlight shape differences.

The metatarsals of KUVP 129724 can be distinguished from diplodocid ones by the absence of a well-developed posterolateral process on the distal articular surfaces of mt I and II, and from flagellicaudatan metatarsals more generally by the lack of distinct rugose ridges close to the dorsolateral edges (McIntosh, Coombs & Russell, 1992; Harris, 2007; Whitlock, 2011; Tschopp, Mateus & Benson, 2015).

Metatarsal I (Fig. 2) has a subrectangular to D-shaped proximal articular surface, with a concave lateral and a convex medial edge. The surface is dorsoplantarly higher than transversely wide. The dorsolateral corner of the proximal articular surface bears a distinct, tapered projection, as occurs in the mt I of the early brachiosaurid Vouivria (Fig. 4; Mannion, Allain & Moine, 2017). The proximal articular surface is strongly beveled compared to the long axis of the shaft, whereas the distal articular surface is approximately perpendicular to it. The distal articular surface is usually similarly beveled as the proximal one in flagellicaudatans (Fig. 4; Janensch, 1961: Beilagen P, R; McIntosh, Coombs & Russell, 1992: Fig. 3; Harris, 2007: Fig. 8; Tschopp & Mateus, 2017: Fig. 75).

Metatarsal II (Fig. 2) is slightly longer than mt I (Table 1). It has a subtrapezoid proximal articular surface with an expanded dorsolateral corner. Both the medial and the lateral edges are dorsoplantarly straight in proximal view (Fig. 5A). As such, it differs from many diplodocids, in which medial and lateral edges are concave (Tschopp, Mateus & Benson, 2015; Tschopp & Mateus, 2017), as well as from the rather subquadrangular shape of the proximal articular surface of mt II in Camarasaurus (Fig. 5A; Tschopp et al., 2015). It most resembles the proximal outline of mt II of Giraffatitan brancai (Fig. 5A), although these also have slightly concave medial and lateral edges (Janensch, 1961; MB.R.2268, E Tschopp, pers. obs., 2014). The shaft of mt II of KUVP 129724 is stout, but less so than in mt I.

Figure 5 Comparative outline drawings of macronarian metatarsals II in proximal (A–C) and dorsal view (D–F).

KUVP 129724 (A, D; left metatarsal) is compared with Giraffatitan MB.R.2181 (B, E; left metatarsal; traced from Janensch (1961)) and Camarasaurus GMNH-PV 101 (C, F; right metatarsal reversed; traced from McIntosh, 1997). Drawings scaled to equal dorsoplantar height (A–C) and proximodistal length (D–F) in order to highlight shape differences.

Metatarsal III (Fig. 2) is the most slender and longest of the five elements (Table 1). The proximal articular surface was damaged during excavation. What remains of the proximal articular surface indicates that the surface had a rhomboid to slightly sheared subrectangular outline, probably similar to Ligabuesaurus (D’Emic, Wilson & Williamson, 2011). It is dorsoplantarly higher than transversely wide. The shaft expands considerably transversely towards the proximal and distal ends. The dorsal surface of the shaft is relatively flat and straight, whereas the plantar surface is concave in lateral view. The distal articular surface has distinct medial and lateral condyles.

Metatarsal IV (Fig. 2) is slightly more robust than mt III. It has a subtriangular proximal articular surface (Fig. 6A), which is different from the L-shaped one of Camarasaurus (Fig. 6A; Tschopp et al., 2015), and the kidney-shaped surface of the putative brachiosaurid Europasaurus (Fig. 6A; DFMMh FV886.3; E Tschopp, pers. obs., 2014). The distal articular surface is beveled medially, so that the medial side of the bone is shorter than the lateral one. Such a beveling has been identified as a synapomorphy for Brachiosauridae by D’Emic (2012) and Mannion et al. (2013).

Figure 6 Comparative outline drawings of macronarian metatarsals IV in proximal (A–C) and dorsal view (D–F).

KUVP 129724 (A, D; left metatarsal) is compared with Camarasaurus SMA 0002 (B, E; right metatarsal reversed; traced from Tschopp et al., 2015) and Europasaurus DFMMh-FV886-3 (C, F; right metatarsal reversed; traced from photo by E Tschopp from 2014). Drawings scaled to equal dorsoplantar height (A–C) and proximodistal length (D–F) in order to highlight shape differences.

Metatarsal V (Fig. 2) has a widely expanded proximal end, which strongly tapers into a long slender shaft, similar to the brachiosaurids Giraffatitan brancai (Janensch, 1961) and Sonorasaurus (D’Emic, Foreman & Jud, 2016). In Janenschia and Camarasaurus, the expansion is wide too, but it extends further distally along the shaft (Fig. 7; Bonaparte, Heinrich & Wild, 2000; Tschopp et al., 2015), whereas in many diplodocids, the proximal expansion is similarly developed as in KUVP 129724 (Fig. 7; Janensch, 1961; Tschopp & Mateus, 2017). The distal articular surface of mt V of KUVP 129724 is only weakly transversely expanded compared to minimum shaft width, which is similar to Camarasaurus, but different from flagellicaudatans (Janensch, 1961; Remes, 2009; Tschopp et al., 2015; Tschopp & Mateus, 2017), see Table S1 and Fig. 7 for mt V proportions). The distal articular surface of mt V of KUVP 129724 is less expanded in relation to proximodistal length than the metatarsals V of both Camarasaurus and diplodocids, and are instead comparable to the somphospondylians Tastavinsaurus and MUCPv-1533 (Canudo, Royo-Torres & Cuenca-Bescós, 2008; González Riga, Calvo & Porfiri, 2008) and the brachiosaurids Cedarosaurus and Sonorasaurus (Fig. 7; D’Emic, 2013; D’Emic, Foreman & Jud, 2016).

Figure 7 Shape differences in sauropod metatarsals V.

The graph represents morphospace occupation of sauropod mt V when comparing proximal transverse widths (prW; x-axis) and distal transverse widths (diW; y-axis) with proximodistal length (pdL). The left mt V of KUVP 129724 is within the morphospace occupied by titanosauriform sauropods (Brachiosauridae + Somphospondyli), and clearly outside non-titanosauriform macronarians like Camarasaurus and Janenschia. Measurements and sources are provided as Table S1. Outlines of selected specimens are traced from the following publications: Janenschia robusta SMNS 12144 (right reversed) from a photo taken by J Nair in 2014, Camarasaurus sp. SMA 0002 (right reversed) from Tschopp et al. (2015), Omeisaurus tianfuensis ZDM T5704 (left) from He, Li & Cai, 1988, Tastavinsaurus sanzi MPZ 99/9 (right reversed); traced from Canudo, Royo-Torres & Cuenca-Bescós (2008), Sonorasaurus thompsoni ASDM 500 (right reversed) from D’Emic, Foreman & Jud (2016), Cedarosaurus weiskopfae DMNH 39045 (right reversed) from D’Emic (2013), and the indeterminate diplodocid MB.R.2371 (left) from a photo taken by E Tschopp in 2014. The metatarsals are scaled to equal proximodistal length to highlight shape differences.

Pedal phalanges

The phalanges (Fig. 2) are generally wider than long (Table 1) and have distinctly expanded proximal articular surfaces and no collateral ligament pits, which is typical for eusauropods (Upchurch, 1998; Wilson & Sereno, 1998; Wilson, 2002; Upchurch, Barrett & Dodson, 2004). In php II-1, III-1, and IV-1, also the distal articular surfaces are expanded transversely.

Phalanx php I-1 (Fig. 2) is just slightly wider than dorsoplantarly high, both proximally and distally, resembling the proportions of Giraffatitan (Janensch, 1961) and diplodocids (Tschopp & Mateus, 2017), but not Camarasaurus (Tschopp et al., 2015). The proximal articular surface lacks the plantar “lip” typical for diplodocids (Upchurch, Tomida & Barrett, 2004; Whitlock, 2011; Tschopp, Mateus & Benson, 2015). The distal articular surface projects slightly dorsomedially, resulting in a distinctly concave medial edge. This corner is equally developed in Giraffatitan (Janensch, 1961) and Sonorasaurus (D’Emic, Foreman & Jud, 2016), but no projection occurs in any other sauropod taxon known to us (Fig. 8).

Figure 8 Comparative outline drawings of macronarian pedal phalanges I-1 in lateral and distal view.

KUVP 129724 (left) is compared with the brachiosaurids Sonorasaurus (ASDM 500, right reversed; traced from D’Emic, Foreman & Jud, 2016), and Giraffatitan (MB.R.2287, left; Janensch, 1961), and the camarasaurid Camarasaurus (SMA 0002, right reversed; traced from Tschopp et al., 2015). Note the straight to concave medial margin of the distal articular surface in the brachiosaurid phalanges, and their elongated shape in lateral view. No lateral view was available from Giraffatitan. Drawings scaled to equal dorsoplantar height in order to highlight shape differences.

The putative php II-1 and III-1 of KUVP 129724 (Fig. 2) are relatively short, compared to Giraffatitan (Janensch, 1961), and more similar in proportion to Camarasaurus (Tschopp et al., 2015). However, the distal condyles of php III-1 of KUVP 129724 are less distinct in dorsal view than in Camarasaurus (Tschopp et al., 2015), and resemble more the state in Giraffatitan (Janensch, 1961).

The other two non-ungual phalanges do not provide any particular morphological information for comparative purposes. Phalanx IV-1 has a very irregular dorsal surface (Fig. 2). The smallest element (KUVP 133862) is a nubbin-like bone typical for the reduced terminal, non-ungual phalanges of digits IV and V of most neosauropods (Bonnan, 2005).

Pedal unguals

Two unguals were recovered with the pedal elements (Fig. 2). The larger of the two (KUVP 144767; interpreted to be php I-2 herein) has the typical sickle-shape of eusauropod unguals (Wilson & Sereno, 1998), whereas the smaller ungual (part of KUVP 129724; interpreted to be php III-4) is rather straight (Fig. 9). The high dorsal projection of the proximal articular surface is however also present in Giraffatitan (Janensch, 1961) and Sonorasaurus (D’Emic, Foreman & Jud, 2016). The proximal and distal outlines resemble Giraffatitan (Janensch, 1961). The scalene cross-section of the unguals differs from the isosceles shape of Camarasaurus (Fig. 9; Tschopp et al., 2015).

Figure 9 Comparative outline drawings of macronarian pedal unguals I and III in lateral view.

KUVP 129724 and 144767 (left) are compared with the brachiosaurids Sonorasaurus (ASDM 500, right reversed; traced from D’Emic, Foreman & Jud, 2016), and Giraffatitan (MB.R. XX 2, left; Janensch, 1961), and the camarasaurid Camarasaurus (SMA 0002, right reversed; traced from Tschopp et al., 2015). No ungual III is known from Sonorasaurus and Giraffatitan. Drawings of unguals I scaled to equal dorsoplantar height in order to highlight shape differences; drawing of unguals III are scaled proportionally to their respective ungual I to show relative sizes of the unguals in the pedes of the included taxa.

Discussion

Systematics

The morphological comparisons lead to an identification of the pes as belonging to Titanosauriformes, and more specifically Brachiosauridae, in particular due to the orientation of the distal articular surface of mt IV that was recovered as a synapomorphy for the clade in two independent phylogenetic analyses (D’Emic, 2012; Mannion et al., 2013). In addition, the elongation of mt V is most similar to titanosauriform taxa sampled herein (see Fig. 7 and Table S1); Camarasaurus has more widely expanded proximal and distal articular surfaces relative to proximodistal length, whereas diplodocids all have more widely expanded distal articular surfaces. The morphology of the phalanx php I-1, with its rounded proximal articular surface and the dorsomedial projection on the distal articular surface strongly suggest a close affinity with the brachiosaurids Giraffatitan and Sonorasaurus. Finally, the relatively straight ungual php III-3 of KUVP 129724 resembles the latter two taxa most and its scalene triangle cross section differs substantially from the isosceles triangle cross section of Camarasaurus KUVP 129716 (A Maltese, pers. obs., 2018). This shape rarely occurs outside of Brachiosauridae. The features distinguishing KUVP 129724 from Giraffatitan are most likely representing differences at a lower taxonomic level within Brachiosauridae, given that many of them are more variable among eusauropods than the traits mentioned above.

The only currently known titanosauriform taxon from the Morrison Formation is Brachiosaurus altithorax. The type locality for this species is close to the town of Grand Junction, Colorado (Riggs, 1903; Riggs, 1904; Fig. 3), and several other localities have been reported to have produced brachiosaurid material in the meantime (Jensen, 1987; Curtice, Stadtman & Curtice, 1996; Carpenter & Tidwell, 1998; Bonnan & Wedel, 2004; Taylor, 2009; Bader, Hasiotis & Martin, 2009). However, the absolute number of brachiosaurid specimens from the Morrison Formation is still low relative to other sauropods, and none of these specimens preserve any bones from the lower hindleg (Taylor, 2009), so that no overlapping material of Brachiosaurus exists with which the pes described herein could be compared. Therefore, even though attribution to Brachiosaurus seems reasonable, we cautiously refer KUVP 129724, 133862, 142200, and KUVP 144767 to Brachiosauridae indet.

Table 2 Sauropod metatarsal proximodistal lengths of the largest specimens (to our knowledge) of selected species (in mm).

Ordered after size within major sauropod subclades. Asterisks mark estimated measurements. Specimen numbers and left (L) and right (R) pedes are indicated, and specified with the single measurements where metatarsals of a single pes have different specimen numbers.

Non-neosauropod Eusauropoda	
	Turiasaurus	Jobaria	Omeisaurus	Cetiosauriscus	Omeisaurus	Shunosaurus	
	riodevensis	tiguidensis	tianfuensis	stewarti	tianfuensis	lii	
	CPT; L	MNN TIG4	ZDM T5704; R	NHMUK R3078; L	ZDM T5701; L	ZDM T5402; L	
Metatarsal I	230 (CPT-1318)		165	152	192	110	
Metatarsal II	300 (CPT-1309)		215	204	202	150	
Metatarsal III	300 (CPT-3967)	300		212		180	
Metatarsal IV	280 (CPT-1268)			207			
Metatarsal V	 245 (CPT-3965)			187			
Source	R Royo-Torres, pers. comm., 2018	Sereno et al. (1999)	He, Li & Cai (1988)	E Tschopp, pers. obs., 2011	He, Li & Cai (1988)	Zhang (1988)	
Diplodocoidea	
 	?Barosaurus	Apatosaurus	Diplodocus	Suuwassea	Galeamopus	Dyslocosaurus	
	lentus	louisae	carnegii	emilieae	pabsti	polyonychius	
	?CM 11984; L	CM 3018; L	CM 94; L	ANS 21122; R	SMA 0011; L	AC 663; L	
Metatarsal I	208	195	163	130.7	124	123	
Metatarsal II	217	213	191	154.3	153	140	
Metatarsal III	242	236	213		164	171	
Metatarsal IV	239	236	206	172.8	180		
Metatarsal V	231		160		178		
Source	McIntosh (2005)	Gilmore (1936)	Hatcher (1901); Mazzetta, Christiansen & Fariña (2004)	Harris (2007)	Tschopp & Mateus (2017)	McIntosh, Coombs & Russell (1992)	
Non-titanosauriform Macronaria	
 	Camarasaurus	Camarasaurus	Camarasaurus	Janenschia	Camarasaurus	Camarasaurus	
	supremus	grandis	grandis	robusta	sp.	lentus	
	AMNH FARB 5761; R	GMNH-PV 101; R	YPM VP.001905; L	SMNS 12144; R	SMA 0002; R	CM 11338; L	
Metatarsal I		172	133	140	113	70	
Metatarsal II		193	174	160	134	90	
Metatarsal III	225	223	182	160	133	88	
Metatarsal IV		206	165	150	112	80	
Metatarsal V		166	125	115	108	60	
Source	Osborn & Mook (1921)	McIntosh et al. (1996)	E Tschopp & O Mateus, pers. obs., 2014	Fraas (1908); J Nair, pers. comm., 2015	Tschopp et al. (2015)	Gilmore (1925)	
Brachiosauridae	
 	Brachiosauridae	Giraffatitan	Sonorasaurus	Vouivria	Cedarosaurus	Venenosaurus	
	indet.	brancai	thompsoni	damparisensis	weiskopfae	dicrocei	
	KUVP 129724; L	MB.R.2181	ASDM 500; R	MNHN.F.1934.6; L	DMNS 39045;	DMNS 40932; R	
Metatarsal I	266		194	175	165	128	
Metatarsal II	290	276	242		205		
Metatarsal III	332			234		172	
Metatarsal IV	329		261		247	180	
Metatarsal V	269		221				
Source	This study	Paul (1988)	D’Emic, Foreman & Jud (2016)	Mannion, Allain & Moine (2017)	A Maltese, pers. obs., 2012	A Maltese, pers. obs., 2012	
Somphospondyli	
	Dreadnoughtus	Alamosaurus	Tastavinsaurus	Ligabuesaurus	Notocolossus	Opisthocoelicaudia	
	schrani	sanjuanensis	sanzi	leanzai	gonzalezparejasi	skarzynskii	
	MPM-PV 1156; R	NMMNH P-49967; R	MPZ 99/9; R	MCF-PHV-233; R	UNCUYO-LD 302; R	ZPAL MgD-I/48; R	
Metatarsal I	210	195	162	140	164	150	
Metatarsal II	250	245	190	190	185	180	
Metatarsal III		270	230	220	197	200	
Metatarsal IV		291	212	220	218	180	
Metatarsal V		281	180	180	196	140	
Source	Lacovara et al. (2014)	D’Emic, Wilson & Williamson (2011)	Canudo, Royo-Torres & Cuenca-Bescós (2008)	Bonaparte, Riga & Apesteguía (2006)	González Riga et al. (2016)	Borsuk-Bialynicka (1977)	

The largest neosauropod pes

Although the taxonomic position of the new specimen cannot be determined for certain, it does represent a dinosaur of enormous proportions. Indeed, the metatarsals of KUVP 129724 are slightly larger than the largest ones of Giraffatitan, and they are considerably larger than those of Dreadnoughtus, which was reported to be one of the largest sauropods ever found (Table 2; Lacovara et al., 2014). The only other sauropod pes known so far that is close to these proportions is from the non-neosauropod eusauropod Turiasaurus riodevensis from the Late Jurassic of Spain (Royo-Torres, Cobos & Alcalá, 2006; R Royo-Torres, pers. commm., 2018).

Based on the hindlimb proportions of the brachiosaurid Vouivria (Mannion, Allain & Moine, 2017), we estimated a femur length of 2071 mm and a tibia length of 1,220 mm for KUVP 129724. This is slightly larger (2%) than the type specimen of Brachiosaurus altithorax (2,030 mm femur length; Riggs, 1903). Assuming that the cartilage caps on the proximal and distal articular surfaces of the longbones would increase their length by approximately 10% (Schwarz, Wings & Meyer, 2007; Bonnan et al., 2010; Holliday et al., 2010), this would result in a hip height of approximately 3.99 m. Although this appears to be the largest pes reported to date, traces and other incomplete body fossils show that the pes described herein does not represent the maximum body size of sauropod dinosaurs. Some of the largest sauropods such as Argentinosaurus or Patagotitan do not preserve pedal material but have femur lengths that considerably exceed our estimate for KUVP 129724 (Argentinosaurus: 2,557 mm, estimated based on incomplete femur; Patagotitan: 2,360 mm; Mazzetta, Christiansen & Fariña, 2004; Carballido et al., 2017). The largest sauropod tracks from the Broome Sandstone of Australia are >1,100 mm in diameter, indicating a similar hip height as calculated for KUVP 129724 herein (>3.41 m; Salisbury et al., 2016). However, all these finds are from the Cretaceous, so that the type specimen of Brachiosaurus altithorax and the pedal elements described herein still represent the largest individual specimens found in the Morrison Formation, only matched in size during the same period by Turiasaurus from Spain and Giraffatitan from Tanzania. Given that the type specimen of Brachiosaurus altithorax was found in western Colorado (Riggs, 1904) and the pes described herein in northeastern Wyoming, this shows that sauropods with very large body size were distributed across wide ranges in the Morrison Formation.

Brachiosaurid distribution in the Late Jurassic of North America

Our detailed description and systematic assessment of the pedal elements KUVP 129724, 133862, 142200, and 144767 confirms the presence of large-sized brachiosaurids in the Upper Jurassic Morrison Formation of the Black Hills. Together with the small-sized brachiosaur mentioned in Bader, Hasiotis & Martin (2009), this pes is the northern-most occurrence of this taxon reported so far in the Late Jurassic of North America (Fig. 10). If the material described herein belonged to the currently only known Late Jurassic North American species Brachiosaurus altithorax, this taxon would cover a range of latitudes across the Morrison Formation. Brachiosaurids, like camarasaurids, were sauropods with broad-crowned teeth, which could process relatively tougher vegetation than the peg-like diplodocoid teeth (Janensch, 1935; Calvo, 1994; Wiersma & Sander, 2017). It would, therefore, seem reasonable to assume they could cover a wide range of vegetational zones. Camarasaurids are also known to (seasonally) migrate (Fricke, Hencecroth & Hoerner, 2011), and Camarasaurus specimens have been found from New Mexico to Montana (Ikejiri, 2005; Woodruff & Foster, 2017). Given the similarities in tooth crown morphology in the two genera, brachiosaurs could have displayed similar geographical spreading and/or migrational habits as camarasaurids. However, additional information will be needed to assess species diversity within brachiosaurids of the Morrison Formation, and to understand in more detail how their distribution, the climate, and vegetation changed throughout the time of deposition of the formation. This is outside of the scope of the current study.

Figure 10 Map of occurrences of Brachiosauridae in the Upper Jurassic Morrison Formation.

The locality of the pes described herein (1) and the type locality of Brachiosaurus altithorax (5) are highlighted in red. The gray area indicates the distribution of the Morrison Formation. 1, Bobcat Pit, Weston County, WY; 2, Freezeout Hills general, Carbon Co., WY; 3, Reed’s Quarry 13, Albany Co., WY; 4, Jensen/Jensen Quarry, Uintah Co., UT; 5, Fruita Paleontological Area general, Mesa Co., CO; 6, Riggs Quarry 13, Mesa Co., CO; 7, Dry Mesa Quarry, Mesa Co., CO; 8, Potter Creek Quarry, Montrose Co., CO; 9, Felch Quarry 1, Fremont Co., CO; 10, Kenton Pit 1, Cimarron Co., OK. Modified from Bonnan & Wedel (2004: fig. 2).

Conclusion

We present the first brachiosaurid pedal elements from the Late Jurassic of North America. The pes represents the largest sauropod pes described to date. Size estimations scaled due to lack of anatomical overlap indicate that these pedal elements belonged to a brachiosaur slightly larger than the holotype of Brachiosaurus altithorax. Moreover, this pes and a small specimen of a brachiosaur from the same quarry represent the northernmost occurrences of the taxon in the Morrison Formation.

List of Institutional Abbreviations

AC Beneski Museum of Natural History of the Amherst College, Amherst, Massachusetts, USA

AMNH American Museum of Natural History, New York City, New York, USA

ANS Academy of Natural Sciences, Philadelphia, Pennsylvania, USA

ASDM Arizona-Sonora Desert Museum, Tucson, Arizona, USA

CM Carnegie Museum of Natural History, Pittsburgh, Pennsylvania, USA

CMC-PV Cincinnati Museum of Natural History and Science, Cincinnati, OH

CPT Museo de la Fundación Conjunto Paleontológico de Teruel-Dinópolis, Aragón, Spain

DFMMh Dinosaurier-Freilichtmuseum, Münchehagen, Germany

DMNS Denver Museum of Nature and Science, Denver, Colorado, USA (previously DMNH)

FMNH Field Museum of Natural History, Chicago, Illinois, USA

GMNH-PV Gunma Museum of Natural History, Gunma, Japan

IANIGLA-PV Instituto Argentino de Nivologia Glaciologia y Ciencias Ambientales, Mendoza, Argentina

KUVP Kansas University Museum of Natural History, Vertebrate Paleontology, Lawrence, Kansas, USA

Mal Malawi Department of Antiquities Collection, Lilongwe and Nguludi, Malawi

MBR Museum für Naturkunde Berlin, Berlin, Germany

MCF-PHV Museo ‘Carmen Funes’, Plaza Huincul, Neuquén, Argentina

MNHN Muséum National d’Histoire Naturelle, Paris, France

MNN Musee National du Niger, Niamey, Republic of Niger

MPCA-PV Colección de Paleovertebrados de la Museum Provincial de Cipolletti ‘Carlos Ameghino’, Cipolletti, Rio Negro Province, Argentina

MPM Museo Padre Molina, Rio Gallegos, Santa Cruz, Argentina

MPZ Museo Paleontológico de Zaragoza, Zaragoza, Spain

MUCPv Museo de Geología y Paleontología Universidad Nacional de Comahue, Argentina

NHMUK Natural History Museum, London, United Kingdom

NMMNH New Mexico Museum of Natural History and Science, Albuquerque, New Mexico, USA

NSMT National Science Museum, Tokyo, Japan

SMA Sauriermuseum Aathal, Switzerland

SMNS Staatliches Museum für Naturkunde, Stuttgart, Germany

UNCUYO-LD Universidad Nacional de Cuyo, Laboratorio de Dinosaurios, Mendoza, Argentina

UNPSJB-PV Universidad Nacional de la Patagonia San Juan Bosco, Comodoro Rivadavia, Argentina

YPM Yale Peabody Museum of Natural History, New Haven, Connecticut, USA

ZDM Zigong Dinosaur Museum, Zigong, China

ZPAL Institute of Paleobiology, Polish Academy of Sciences, Warsaw, Poland

Supplemental Information

Table S1 Eusauropod metatarsal V elongation

Red ratios are based on measurements taken from figures.

Click here for additional data file.

The pes described herein was found during an expedition spearheaded by the late Larry Martin of KUVP. K Bader, J Richard and M Christopher prepared the individual bones. The late Jack McIntosh first proposed this pes could belong to Brachiosaurus. We thank the following people for access to comparative material: Nils Knötschke (DFMMh), Daniela Schwarz (MB.R.), Paul Barrett and Sandra Chapman (NHMUK), Dan Brinkman and Jacques Gauthier (YPM). Mike Triebold generously provided the 3D scanning and rendering equipment for the figures. Chris Beard is thanked for enabling scanning at KUVP. A special thanks goes to Matthew Christopher for his expert aid in manipulating the 3D models for figures. Jay Nair shared photographs and measurements of Janenschia. We are grateful to editor Matthew Wedel, reviewer Mike Taylor, and two anonymous reviewers, whose comments and helpful suggestions greatly improved this paper.

Additional Information and Declarations

Competing Interests

Author Contributions

Data Availability

Anthony Maltese is an employee of the Rocky Mountain Dinosaur Resource Center.

Anthony Maltese conceived and designed the experiments, performed the experiments, analyzed the data, contributed reagents/materials/analysis tools, prepared figures and/or tables, authored or reviewed drafts of the paper, approved the final draft.

Emanuel Tschopp performed the experiments, analyzed the data, contributed reagents/materials/analysis tools, prepared figures and/or tables, authored or reviewed drafts of the paper, approved the final draft.

Femke Holwerda performed the experiments, analyzed the data, prepared figures and/or tables, authored or reviewed drafts of the paper, approved the final draft.

David Burnham conceived and designed the experiments, performed the experiments, analyzed the data, contributed reagents/materials/analysis tools, approved the final draft.

The following information was supplied regarding data availability:

The raw measurements for Fig. 7 are provided in Table S1. The remaining raw data are included in the article (measurements in tables, anatomical data in descriptions).

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
