# Peer review of "The real Bigfoot: a pes from Wyoming, USA is the largest sauropod pes ever reported and the northern-most occurrence of brachiosaurids in the Upper Jurassic Morrison Formation"

_PeerJ, doi:10.7717/peerj.5250_

## Round 0.1 · original submission · Minor Revisions

Congratulations - all three reviewers found value in your manuscript, and their comments are constructive.

One point that I don't think was raised by the reviewers: in both the abstract and the conclusion, you refer to these specimens as the first brachiosaurid pedal elements from the United States (abstract) or North America (conclusion). This cannot be true if Sonorasaurus is a brachiosaurid, which is both how it has been coming out in the most recent phylogenetic analyses, and how you refer to it yourself in the text. They might be the first pedal elements of a Morrison or North American Jurassic brachiosaurid, but they can't be the first North American brachiosaurid pedal elements full stop, unless Sonorasaurus is not a brachiosaurid. You may wish to discuss the phylogenetic position of Sonorasaurus in the revised manuscript, or modify the claim.

Also, I think it is important to clarify that this may be the largest neosauropod pes recovered to date, but both tracks (e.g., Broome Sandstone) and body fossils of larger sauropods for which pedes have not been recovered yet (Argentinosaurus, Patagotitan, etc.) suggest that the largest sauropod pedes were significantly larger still. Precisely because claims of large dinosaurs tend to drive media coverage of our field, it's important to distinguish between "largest found to date" and "largest that ever existed".

In revising the manuscript and drafting your response letter, please be diligent in addressing all of the issues raised by the reviewers - I've reviewed them all and find them worthy of engaging with. In particular, I agree with the point raised by the reviewers that the manuscript would be greatly improved by the addition of comparative figures that show explicitly how these elements differ from those of contemporaneous diplodocids and especially Camarasaurus.

I look forward to seeing an improved version of this work soon.

·

Basic reporting

[***NOTE*** I see that copy-pasting this review has destroyed all the formatting. I suggest the authors read the original version, which I have uploaded to http://www.miketaylor.org.uk/tmp/secret/Bigfoot-comments-MikeTaylor.doc and which contains exactly the same information as I have pasted here, but more legibly.]

This is a good paper providing a clear description and adequate illustrations of a significant specimen. The paper can be published in essentially its present form, although I list below some minor points that should be improved. I also suggest some further additions that the authors may wish to make, but which should not be required for publication.
The writing is clear and unambiguous throughout, with the exception of a very few phrases which I mention below. The introduction establishes the context of this work well.
References to the literature and plentiful and relevant. However, in many cases, citing page numbers would be a real improvement. For example in lines 150–132, the manuscript reads “The proximal and distal articular surfaces [of metatarsal I] are slightly beveled compared to the long axis of the shaft, but not to the degree as seen in flagellicaudatans (Janensch, 1961; McIntosh, Coombs & Russell, 1992; Harris, 2007; Whitlock, 2011; Tschopp, Mateus & Benson, 2015; Tschopp & Mateus, 2017).” Especially when referring to long papers such as that of Janensch, which runs to 59 pages, it would make a big difference to provide page numbers, so the reader can go straight to the relevant section. That goes double for Tschopp and Mateus 2017!
Some specifics:
The claim that “this [pes] represents the northernmost occurrence of a brachiosaurid in the Morrison Formation” cannot be strictly correct given the presence of “a nearly complete small brachiosaur” in the same quarry, and in light of the fact that “Brachiosaurid material from this site has been reported in the past (Foster, 2003; Bader, Hasiotis & Martin, 2009)”. This should be easy to reword, both in the abstract and in the places where similar claims occur in the main text. It may be wise to amend the title, too – either to make the northernmost claim more precise, or to excise it for a shorter title.
“The pes described herein consists of an astragalus (KUVP 142200), metatarsals I to V, five non-ungual pedal phalanges, and one ungual (KUVP 129724)”. It would be useful to add these KUVP numbers to Table 1, and explain how they differ in meaning from the field numbers.
Within the “Description and Comparison” section, the authors may wish to consider adding sub-headings for the astragalus, metatarsals, phalanges and ungual.
“The astragalus KUVP 142200 (Figure 2A) is slightly wider transversely than proximodistally tall and anteroposteriorly long.” It would be useful to explain how the orientation of this element was established. It would also be helpful to have this element illustrated in more views – perhaps all six cardinals. (I assume the necessary photos already exist.)
“The lateral surface of the astragalus KUVP 142200 received the fibula. It faces laterally, and has no distinct bony shelf that would have supported the fibula, unlike the condition in diplodocids.” It would be helpful to include comparative illustrations here and elsewhere: one of the joys of born-OA venues like PeerJ is that there is no real limit on what illustrations can be included, so it’s often good to take advantage of this to help the reader. Similarly, “The metatarsals of KUVP 129724 can be distinguished from diplodocid ones by the absence of a well-developed posterolateral process on the distal articular surfaces of mt I and II, and from flagellicaudatan metatarsals more generally by the lack of distinct rugose ridges close to the dorsolateral edges”: this would be easier for non-pes specialists to makes sense of if comparative diagrams were included. (That said, this is only a suggestion, not a requirement! If it’s inconvenient to do, just skip it and get this paper out. I don’t want to be That Guy.)
“The dorsolateral corner of the proximal articular surface bears a distinct, tapered projection.” I couldn’t locate this in the illustration: please highlight it.
“The proximal and distal articular surfaces are slightly beveled compared to the long axis of the shaft.” It would be helpful to see this in lateral views of the metatarsals. Again, there is the option of including many more illustrations – why not take advantage of it?
“The distal articular surface is beveled medially.” What does this mean? That the medial aspect of the distal articular surface is located more proximally than the lateral aspect? Or vice versa?
“The distal articular surface [of phalanx I-1] projects slightly dorsomedially.” I can’t locate this projection: please highlight it in the illustration.
“This is slightly larger than the type specimen of Brachiosaurus altithorax (2030 mm femur length; Riggs, 1904).” I would move this fact up ahead of the speculation about cartilage thickness. Also perhaps specify that it’s only 2% larger – which is not to be sniffed at, but not quite as spectacular as people might imagine otherwise.
“Our detailed description and systematic assessment of the pes KUVP 129724 confirmed the presence of large-sized brachiosaurids.” It would be more usual to say “confirms”.
The abbreviations “mtt” is used only in Table 1, where space is not at a premium, and “php” is similarly rarely used. It might be better to write the terms out in full: “metatarsal” and “phalax”. (There’s no need to specify “pedal”, as this paper doesn’t mention hands.) In any case, isn’t “mt” the usual abbreviation for “metatarsal”? It’s used elsewhere in the manuscript, e.g. lines 133ff.
Two of the measurements in Table 1 (proximal width of mtt III and distal width of mtt IV) carry asterisks, but their meaning is not explained in the table caption.
The field numbers for the astragalus and phalanx V-1 are six-digit numbers, while those of all the other elements are three-digit numbers with a “BP” prefix. Why the discrepancy? It might be useful to discuss the history of the assignment in the text. Oddly, these are not the same two elements that have their own KUVP numbers. Perhaps discuss the assignment of the various numbers in more detail.
Table 2 is interesting. It would be helpful to explain a little more in the caption – for example, what the various rows contain, especially the “L” and “R” that sometimes appear after semicolons. (Yes, I figured out that they mean “left” and “right”, but it helps to be explicit.)
Figure 1 is excellent, and very helpful in understanding how the elements were found. However, it would be more useful still if the four parts were designated A–D, and citations added to each part where relevant in the text. It would also be helpful to add specimen numbers to the quarry maps, including the identities of the three partial skeletons that are shown.
In Figure 2, does the scale bar apply to all parts of the illustration (e.g. including the astragalus)?
More importantly, what is the orientation of the various elements in Figure 2? In particular, I can’t be sure whether in the proximal views of the metatarsals, the dorsal surface is at the top or the bottom. (A case could be made for composing the illustration either manner.)
In Figure 2, the proximal facet of mt III looks far wider in dorsal view than it does in proximal view. Are the parts of this illustration correctly oriented and scaled?
In the text, phalanges are referred to as II-1, III-1, etc., but in the Figure 2 caption, they are simply called II, III, etc. Please be consistent.
Figure 3 is clear, but rather clumsily executed: for example, the grey shading does not always reach the outline of the Morrison Formation. It might be worth tidying up a bit.

Experimental design

This research is fully within the scope of the journal. However, the other questions that reviewers are asked to address in this section and the next do not really apply to descriptive/comparative papers such as this one. There is no “experiment” whose design can be comment on, and no numerical data whose statistical power can be assessed. But this is a solid and well-constructed paper by the standards of descriptive palaeontology.

Validity of the findings

I am satisfied that the conclusion – that the specimen belongs to Brachiosauridae – is well supported by the evidence that the authors present.

Additional comments

In understanding the morphology that this paper describes, I found it helpful to refer to a photo of an articulated pes of the Berlin Giraffatitan mount. My photo is not publication quality, but if any of the authors have good photos of this mount, it might be worth including one or more images of its left pes (or right pes reversed) in an additional figure for comparative purposes. I know that the elements of the mount’s pedes are sculptures, but they are based on real elements, and adequately represent the form of the complete foot. Once more, this is an optional recommendation, and if the authors do not have such images (or if they simply disagree with me), that’s fine.
Finally, another purely optional suggestion. We now live in a time where it is cheap and easy to create 3D models of bones using photogrammetry: see for example the model of the Xenoposeidon holotype vertebra that can be downloaded from https://figshare.com/articles/Untitled_Item/5605612 or browsed at https://sketchfab.com/models/7f88203e0bbb49a194cb254ab05c4b22. The authors could do a great service to their peers if they published similar photogrammetric surface scans of the bones described in this paper: for example, other researchers could attempt to articulate the bones virtually into a complete pes, model the associated muscles, etc. I do not recommend delaying publication until this is done – I just recommend it to the authors as something well worth doing in parallel with or subsequent to the publication process.

Reviewer 2 ·

Basic reporting

The manuscript is written in clear, unambiguous, professional English language. The introduction provides enough background to be informative and the literature documenting the known distribution and occurrence of Morrison sauropod pedes is appropriate and well-referenced. The structure conforms to Peerj standards and the basic norms for paleontology in general. The figures are excellent, but there should be other figures as I will discuss in the general comments. That said, in particular I really appreciated Figure 1 which does a great job of showing the geographic location and quarry map in an intuitive and clear way. The raw data which I take to be the measurements of the pes are incorporated into the paper.

Experimental design

The research is original and the authors have a pretty simple and straightforward observation that they are reporting the largest sauropod pes ever reported and that this pes expands the northern-most occurrence of brachiosaurids in the Morrison Formation. The investigation and methods are fine, but there should be more substance to this paper as I detail in the general comments.

Validity of the findings

Regarding the validity of the findings, the first part of this research (the largest sauropod pes) is supported by the measurement data – they are reporting the largest sauropod pes ever discovered and described. The inferences about the overall hip height and general dimensions of the hindlimb also follow from their data logically because sauropod hindlimbs are generally similar in proportions and these animal grew isometrically, so their inferences on overall size of the animal are consistent with typical practices in paleontology. Where things get a little less clear and could use some more elaboration is in the second portion of this research where this pes is identified as brachiosaurid, which I will comment on in the general comments.

Additional comments

I first want to commend the authors on reporting and providing photographic documentation of a sauropod pes and its original relation and orientation in a quarry map. Oftentimes such descriptive work is overly simplified or a quarry map is not provided, so this preserves the history of the discovery and will make it possible many years into the future for other paleontologists to understand where this pes came from and how it was associated with other fossils.

I have no issues with the measurements provided or the claim that this is the largest known sauropod pes. It clearly is. What is less clear is whether there is enough data to support this being assigned to a brachiosaurid. It is no doubt the pes of a Macronarian sauropod for all of the reasons the authors articulate, such as the lack of the posterolateral process on metatarsal I. The descriptions of the material are fine, but when it comes to the systematics two things would be incredibly useful and would bolster the authors’ inference that this is a brachiosaurid pes.

First, the comparisons of this pes to other sauropod pedes are all qualitative in nature, and it would be far more robust to do something quantitative, whether that be a phylogenetic analysis wherein the characters of the pes are integrated into a dataset to elucidate where this pes shakes out on the sauropod family tree. If it would be too difficult or not robust enough to run such an analysis, an equally valid approach would be something with morphometrics or even geometric morphometrics to show whether there are statistical differences between the shapes of the pedal elements or selected pedal elements in this sauropod with other known neosauropods. One could run something akin to a canonical variates analysis to see if the shapes or measurements of known neosauropods predict where this pes falls. But something more than simply descriptive comparisons because there is the possibility that this pes is just a very large camarasaurid foot after all. From just a few linear measurements, we cannot tell much beyond the fact that this is a big foot – more quantitative data might show that this cannot be a camarasaurid, which would bolster the authors’ inference. In my experience, non-titanosaur Macronarian pedes can be difficult to tell apart qualitatively.

Second, given all of the comparisons the authors make to other neosauropods and eusauropods generally, more figures are necessary and encouraged so that the reader can see and visualize particular anatomical features and draw their own conclusions. So, for example, when the authors say at line 169 that “Metatarsal V … [is] similar to the brachiosaurids Giraffatitan and Sonorasaurus …” it would be extremely helpful to see that directly in the paper, whether through detailed line drawings, photos, or some combination. I’m glad to see that the authors have photos of nearly all orientations of the pes, but it would also be good to have photos of the ventral in addition to the pictures of the dorsal sides of the metatarsals. Overall, more comparative figures would be extremely helpful and likely further bolster the authors’ inferences that this pes is indeed brachiosaurid.

Overall, the paper provides valuable data on a very large sauropod pes and, if more comparative and quantitative work were done, may expand the range of brachiosaurids in the Morrison.

Reviewer 3 ·

Basic reporting

No comment.

Experimental design

No comment.

Validity of the findings

No comment.

Additional comments

Any chance of including the manus from the KU site near Newcastle? As a second specimen, this would seem relevant to some of the points in the text about range and distribution.

Annotated reviews are not available for download in order to protect the identity of reviewers who chose to remain anonymous.

---

## Round 0.2 · accepted · Accept

Thank you for your diligence in addressing the concerns of the reviewers. I am satisfied with the revised manuscript, and I am happy to accept it for publication in PeerJ.

The decision of whether or not to publish the peer reviews alongside the paper is entirely yours, and will not affect how your paper is handled going forward. However, I encourage you to do so. Making the reviews public allows the reviewers to receive more credit for their efforts, and also contributes to the emerging culture of fairness and transparency in editing and peer review.

#